# Predictive factors involving the remission and recurrence of hypertension post-laparoscopic sleeve gastrectomy in Japanese patients with severe obesity

**Hideki Kumagai** *, **Akira Sasaki, Akira Umemura, Yota Tanahashi, Takafumi Iwasaki, Taro Ando, Hirokatsu Katagiri, Hiroyuki Nitta**

Department of Surgery, School of Medicine, Iwate Medical University, Yahaba, Iwate, Japan

* hishioka@iwate-med.ac.jp

**Data Availability Statement:** All relevant data are within the manuscript and its Supporting Information files.

## Abstract

Metabolic surgery, including laparoscopic sleeve gastrectomy (LSG), may improve hypertension (HTN) complicated by severe obesity; however, insufficient deliberation exists regarding the therapeutic effect of post-metabolic surgery on HTN. This study aimed to analyze the factors correlated with HTN remission and recurrence post-LSG in patients who have severe obesity, and to create a classification system to predict HTN remission and recurrence. Of the 102 patients who underwent LSG at Iwate Medical University Hospital between 2008 and 2020, 62 were enrolled in this study. Factors correlated with HTN remission and recurrence post-LSG were retrospectively analyzed. The HTN remission rate 12-months post-LSG was 40.3%. The remission cohort had a lower number of preoperative antihypertensive drugs (ADs) than that of the non-remission cohort (one and two tablets, respectively; p< 0.001). Additionally, the remission cohort had a statistically significantly smaller visceral fat area (VFA) than that of the non-remission cohort, at all time points. Logistic regression analysis revealed that the number of preoperative ADs and VFA were independent predictors of remission. The HTN recurrence rate 36-months post-LSG was 36.0%. In the recurrence cohort, the number of preoperative ADs and VFA were higher and larger than that in the non-recurrence cohort, respectively. Stratification, based on the number of preoperative ADs and VFA, revealed that the HTN remission and recurrence rates in the sub-cohort with a small number of preoperative ADs and small VFA (100% and 16.7%, respectively) were better than those in the sub-cohort with a large number of preoperative ADs and large VFA (5.3% and 100%, respectively). In Japanese patients with severe obesity, the number of preoperative ADs and VFA were correlated with HTN remission and recurrence post-LSG. Stratification, by combining the number of preoperative ADs and VFA, may be useful in predicting HTN remission and recurrence.

**Funding:** This work was supported by JSPS KAKENHI Grant Numbers 20K11586 (Akira Sasaki), 22K11812 (Akira Umemura), 23K10852 (Akira Sasaki), and 24K20674 (Taro Ando). Research Grant from Keiryokai Grant Number 148 (Akira Umemura), and Research Grant from Chugai Pharmaceutical Co., Ltd. Grant Number CGPS20230519015 (Akira Umemura). The funders had no role in study design, data collection and analysis, decision to publish, or preparation of the manuscript.

**Competing interests:** The authors have declared that no competing interests exist.

## Introduction

Hypertension (HTN) is strongly correlated with severe obesity. Cardiovascular diseases secondary to HTN account for 70% of obesity-related mortalities [1]. Annually, approximately 4 million people die from cardiovascular diseases associated with obesity [2]. Thus, the prevention of cardiovascular diseases in patients with severe obesity is an urgent matter.

Metabolic surgery, including laparoscopic sleeve gastrectomy (LSG), not only has a weight loss effect but may also improve HTN complicated by severe obesity [1]. Factors correlated with HTN remission and recurrence have been deliberated; however, these factors remain controversial [1,3–5]. In particular, only a few studies exist in Japan [6,7]; and the analyses of factors related to HTN remission and recurrence in post-metabolic surgery can provide valuable indicators for predicting the therapeutic effect thereof for HTN in patients with severe obesity.

The aim of the present study to analyze the factors correlated with HTN remission and recurrence post-LSG in patients with severe obesity at our hospital, and to create a classification system to predict HTN remission and recurrence.

## Methods

### Study design and ethical considerations

This retrospective, observational study was conducted at the Iwate Medical University Hospital, in accordance with the Declaration of Helsinki (1964). The study protocol was approved by the Institutional Ethics Committee of Iwate Medical University Hospital (approval number: MH2022-056). Written informed consent was obtained from all enrolled patients. The data were accessed for research purposes between January 23 and 29, 2024.

### Patients

From June 1, 2008, to December 31, 2020, 102 patients underwent LSG at Iwate Medical University Hospital. Of these, 70 were preoperatively diagnosed with HTN. Eight patients were excluded due to insufficient data, and 62 patients who had undergone a 3-year follow-up were enrolled in the study (Fig 1).

Following the Japanese Society of Hypertension Guidelines, HTN was defined as follows [8]: 1) systolic blood pressure $\geq$ 140 mm Hg, 2) diastolic blood pressure $\geq$ 90 mm Hg, and 3) patients who had been diagnosed with HTN preoperatively or were taking oral antihypertensive drugs (ADs). Blood pressure was measured during outpatient consultations. Patients whose first blood pressure measurement revealed a systolic blood pressure of $\geq$ 140 mm Hg or diastolic blood pressure of $\geq$ 90 mm Hg had their blood pressures re-measured after a suitable period of time, and the lower blood pressure measurement was adopted. In this study, HTN refers to essential HTN; secondary HTN, such as Conn's syndrome and Cushing syndrome, was excluded.

Remission was defined as a patient who no longer satisfied any of the above criteria for HTN, and recurrence was defined as a patient who satisfied any of the above criteria for HTN, post-remission.

All patients met the insurance criteria for LSG in accordance with the 2021 consensus statement of the Japanese Society for the Treatment of Obesity, Japan Diabetes Society, and Japan Society for the Study of Obesity [9].

### Data collection

We analyzed the variables of age, sex, weight, body mass index, excess weight loss, total weight loss, the presence or absence of recurrent weight gain, obesity-related diseases, including type

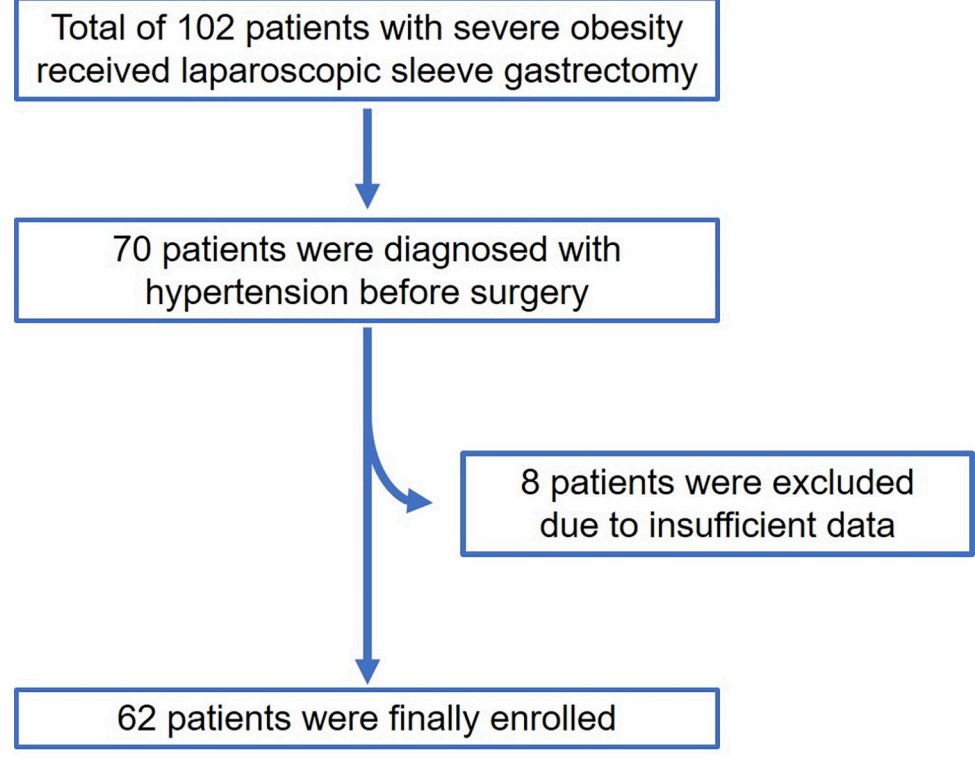

**Fig 1. Flow chart of enrolled patients in this study.**

2 diabetes, obstructive sleep apnea, dyslipidemia, and hyperuricemia, the number of ADs, and the duration of HTN.

In accordance with the International Federation for the Surgery of Obesity and Metabolic Disorders, recurrent weight gain was defined as a weight gain of $> 30\%$ of the initial surgical weight loss or worsening of an obesity-related disease that was a significant indication for surgery [10].

Furthermore, the subcutaneous fat area (SFA), visceral fat area (VFA), plasma renin activity (PRA), plasma aldosterone concentration (PAC), creatinine, hemoglobin A1c (HbA1c), homeostasis model assessment of insulin resistance (HOMA-IR), homeostasis model assessment of beta-cell function (HOMA-β), high-density lipoprotein cholesterol (HDL-C), low-density lipoprotein cholesterol (LDL-C), uric acid, and apnea–hypopnea index (AHI) were evaluated. The SFA and VFA were analyzed using image analysis software (SYNAPSE VINCENT, Fujifilm Medical Co., Ltd., Tokyo, Japan), with computed tomographic (CT) images of the horizontal cross-section at the umbilical level [11]. The VFA reduction rate was calculated as follows: VFA reduction rate = (VFA at baseline − VFA at 6, 12, and 36 months post-LSG) / (VFA at baseline) ×100. The AHI was measured using overnight polysomnography.

## Statistical analyses

Data were presented as medians (interquartile ranges) for continuous variables and as numbers (percentages) for nominal variables. The Mann–Whitney U test was used to compare values between cohorts for continuous variables. The Fisher's exact test was used for the comparison of proportions between the two cohorts for a nominal variable. Univariate and multivariate analyses using logistic regression models were performed to analyze the factors

correlated with HTN remission and recurrence. A receiver operating characteristic (ROC) curve analysis was conducted to evaluate the accuracy of the clinical parameters extracted from the multivariate analysis. Statistical significance was set at $p < 0.05$. All statistical analyses were performed using EZR software version 1.61 (Saitama Medical Center, Jichi Medical University, Saitama, Japan) [12].

## Results

### Comparative analysis of the patient characteristics and clinical parameters between the remission and non-remission cohorts

Of the 62 patients, 25 (40.3%) achieved HTN remission within 12 months post-LSG. Table 1 reveals a comparative analysis of the characteristics and clinical parameters between the remission and non-remission cohorts. The age was significantly younger (41.0- and 47.0-years-old, respectively; p = 0.041). The number of preoperative ADs was statistically significantly lower (one and two tablets, respectively; $p < 0.001$). The duration of HTN was significantly shorter (1 and 10 years, respectively; $p < 0.001$) in the remission cohort than in the non-remission cohort. There was no significant difference in the incidence of obesity-related diseases between the two groups.

The non-remission cohort had a VFA and AHI that were statistically significantly greater at all time points than the remission cohort. Additionally, the VFA reduction rate in the remission cohort was significantly greater at 6 months post-LSG than that in the non-remission cohort (49.8 and 36.4%, respectively; p = 0.004). The remission cohort had better renal function than the non-remission cohort. The effects of weight loss at 6-months and 12 months post-LSG were significantly better in the remission cohort than in the non-remission cohort.

### Univariate and multivariate analysis correlated with HTN remission

Regarding the preoperative factors correlated with HTN remission, age, the number of ADs, duration of HTN, VFA, diastolic blood pressure, and AHI were extracted, using univariate analyses (Table 2). Multivariate analysis using a stepwise method for clinical parameters with a p-value < 0.1 from the univariate analyses revealed that the number of ADs (odds ratio: 0.172, 95% confidence interval [CI]: 0.059–0.505, p = 0.001) and VFA (odds ratio: 0.986, 95% CI: 0.972–0.999, p = 0.040) were independent predictors correlated with HTN remission (Table 3). In the ROC analysis, the area under the curve (AUC) value for the number of ADs was 0.889 (95% CI: 0.806–0.972), the cutoff value was one tablet, the sensitivity was 78.9%, and the specificity was 88.8% (Fig 2A). Conversely, the AUC value for the VFA was 0.694 (95% CI: 0.558–0.830), the cut-off value was 243.0 cm$^2$, the sensitivity was 76.5%, and the specificity was 56.0% (Fig 2B).

### Comparative analysis of the patient characteristics and clinical parameters between the recurrence and non-recurrence cohorts

Of the 25 patients who achieved HTN remission within 12 months post-LSG, 9 (36.0%) were diagnosed with recurrence within 36 months, postoperatively. Table 4 depicts the comparative analysis of the characteristics and clinical parameters between the recurrence and non-recurrence cohorts. The number of preoperative ADs was higher in the recurrence cohort than in the non-recurrence cohort.

Post-LSG, the recurrence cohort had lower and statistically significantly lower, preoperative HbA1c and PRA levels at 12 and 36 months, respectively, than the non-recurrence cohort. The preoperative VFA was higher in the recurrence cohort than in the non-recurrence cohort. Differences in postoperative weight loss between the two cohorts were not observed.

**Table 1. Comparison of patient characteristics and clinical parameters at baseline, 6 months, and 12 months after LSG between the remission and the non-remission groups.**

| Variable | All patients (n = 62) | Remission group (n = 25) | Non-remission group (n = 37) | P value |
|---|---|---|---|---|
| Baseline | | | | |
| Age (years) | 45.0 (36.0–55.0) | 41.0 (33.0–51.0) | 47.0 (41.0–55.0) | **0.041** |
| Gender, n (%) | | | | |
| Male | 35 (55.6) | 11 (44.0) | 23 (63.2) | |
| Female | 28 (44.4) | 14 (56.0) | 14 (36.8) | 0.198 |
| Body weight (kg) | 120.0 (101.5–131.5) | 107.0 (93.0–129.0) | 123.0 (109.0–131.0) | 0.069 |
| Body mass index (kg/m$^2$) | 38.0 (35.0–41.0) | 37.0 (35.0–40.0) | 40.0 (35.0–42.0) | 0.149 |
| Obesity-related diseases, n (%) | | | | |
| Type 2 diabetes | 43 (68.3) | 16 (64.0) | 27 (71.1) | 0.590 |
| Obstructive sleep apnea | 63 (100) | 25 (100) | 38 (100) | − |
| Dyslipidemia | 46 (73.0) | 18 (72.0) | 28 (73.7) | 1.000 |
| Hyperuricemia | 30 (47.6) | 11 (44.0) | 19 (50.0) | 0.797 |
| Antihypertensive drug | 2 (1–2) | 1 (0–1) | 2 (2–3) | **<0.001** |
| Duration of HTN (years) | 8.0 (1.0–13.0) | 1.0 (1.0–11.0) | 10.0 (7.0–14.0) | **<0.001** |
| SFA (cm$^2$) | 494.0 (424.5–584.0) | 525.0 (399.0–621.0) | 485.0 (433.5–548.3) | 0.629 |
| VFA (cm$^2$) | 273.0 (209.0–320.5) | 231.0 (196.0–290.0) | 293.0 (244.3–331.5) | **0.012** |
| Systolic blood pressure (mmHg) | 140.0 (124.5–152.0) | 146.0 (138.0–157.0) | 134.0 (122.0–148.0) | 0.054 |
| Diastolic blood pressure (mmHg) | 84.5 (74.3–92.8) | 91.0 (83.0–95.0) | 80.0 (71.0–90.0) | **0.012** |
| PRA (ng/mL/hr) | 2.4 (1.0–6.0) | 1.2 (0.9–2.9) | 3.6 (1.2–8.4) | 0.070 |
| PAC (pg/mL) | 137.0 (105.5–176.5) | 117.0 (105.8–224.3) | 142.0 (106.5–166.0) | 0.893 |
| Creatinine (mg/dL) | 0.7 (0.6–0.9) | 0.7 (0.6–0.8) | 0.8 (0.7–1.0) | 0.068 |
| HbA1c (%) | 6.8 (5.9–8.1) | 6.7 (5.8–8.1) | 6.9 (5.8–8.1) | 0.561 |
| HOMA-IR (no unit) | 5.0 (3.0–7.0) | 6.0 (3.5–7.5) | 5.0 (2.5–7.0) | 0.370 |
| HOMA-β (no unit) | 108.0 (68.5–222.8) | 127.0 (87.0–235.0) | 103.0 (51.0–201.0) | 0.270 |
| HDL-C (mg/dL) | 41.5 (36.0–48.8) | 43.0 (36.0–49.0) | 40.0 (36.0–48.0) | 0.790 |
| LDL-C (mg/dL) | 115.0 (99.0–136.5) | 112.0 (104.0–152.0) | 109.0 (90.0–126.0) | 0.155 |
| Uric acid (mg/dL) | 6.8 (5.7–7.7) | 6.7 (5.6–7.7) | 6.9 (5.8–7.6) | 0.869 |
| AHI (events/h) | 52.0 (31.3–77.0) | 38.5 (25.0–61.5) | 63.0 (40.8–86.3) | **0.007** |
| 6 months after LSG | | | | |
| Body weight (kg) | 89.0 (74.5–98.0) | 76.0 (71.0–90.0) | 93.0 (83.8–121.0) | **0.003** |
| Body mass index (kg/m$^2$) | 32.0 (29.0–35.0) | 29.0 (28.0–32.0) | 33.0 (30.0–36.0) | **0.002** |
| EWL (%) | 50.0 (42.0–61.0) | 59.0 (48.0–66.0) | 47.0 (40.0–55.0) | **0.003** |
| TWL (%) | 25.0 (21.0–29.0) | 26.0 (23.0–31.0) | 23.0 (20.0–26.8) | 0.051 |
| SFA (cm$^2$) | 494.0 (424.5–584.0) | 310.9 (196.4–420.7) | 326.9 (269.5–374.6) | 0.536 |
| VFA (cm$^2$) | 154.2 (103.8–198.6) | 103.8 (87.5–146.9) | 178.5 (146.8–238.7) | **<0.001** |
| VFA reduction rate (%) | 44.8 (31.4–55.2) | 49.8 (36.5–64.4) | 36.4 (29.4–47.1) | **0.004** |
| Systolic blood pressure (mmHg) | 124.0 (116.5–134.0) | 119.0 (114.0–130.0) | 126.0 (121.0–137.8) | **0.042** |
| Diastolic blood pressure (mmHg) | 76.0 (70.0–83.5) | 74.0 (67.0–81.0) | 77.5 (73.0–90.3) | 0.079 |
| PRA (ng/mL/hr) | 0.9 (0.4–2.3) | 0.7 (0.2–1.3) | 1.4 (0.5–3.0) | 0.111 |
| PAC (pg/mL) | 96.5 (59.8–123.8) | 94.5 (61.5–116.5) | 96.5 (56.8–123.8) | 0.826 |
| Creatinine (mg/dL) | 0.7 (0.6–0.9) | 0.7 (0.6–0.7) | 0.7 (0.6–0.9) | 0.081 |
| HbA1c (%) | 5.5 (5.3–6.1) | 5.4 (4.9–5.5) | 5.9 (5.3–6.3) | **0.009** |
| HOMA-IR (no unit) | 1.4 (0.8–2.2) | 1.1 (0.7–1.9) | 1.8 (1.0–2.2) | 0.194 |
| HOMA-β (no unit) | 91.1 (54.0–136.2) | 87.4 (62.5–189.2) | 93.9 (50.7–133.8) | 0.524 |
| HDL-C (mg/dL) | 51.0 (46.0–59.0) | 50.0 (45.0–58.0) | 52.0 (46.0–59.3) | 0.318 |
| LDL-C (mg/dL) | 108.0 (89.0–129.0) | 120.0 (93.0–130.0) | 103.0 (85.5–125.0) | 0.207 |

*(Continued)*

**Table 1.** (Continued)

| Variable | All patients (n = 62) | Remission group (n = 25) | Non-remission group (n = 37) | P value |
|---|---|---|---|---|
| Uric acid (mg/dL) | 6.2 (5.3–6.8) | 6.0 (5.1–6.6) | 6.3 (5.6–7.3) | 0.189 |
| AHI (events/h) | 20.4 (9.3–42.1) | 11.4 (6.4–19.5) | 41.0 (19.6–53.6) | **<0.001** |
| 12 months after LSG | | | | |
| Body weight (kg) | 86.0 (71.0–94.0) | 72.0 (68.0–88.0) | 89.0 (75.0–96.0) | **0.007** |
| Body mass index (kg/m$^2$) | 30.0 (27.5–34.0) | 28.5 (26.0–31.0) | 32.0 (29.0–36.0) | **0.013** |
| EWL (%) | 56.0 (46.0–67.5) | 62.5 (53.8–73.0) | 53.0 (42.0–62.5) | **0.019** |
| TWL (%) | 26.0 (22.0–31.5) | 29.5 (24.8–32.3) | 24.0 (21.0–31.0) | 0.105 |
| SFA (cm$^2$) | 314.0 (237.8–382.8) | 313.9 (166.8–296.2) | 328.7 (259.5–382.2) | 0.408 |
| VFA (cm$^2$) | 139.5 (93.8–182.6) | 104.3 (69.8–145.3) | 156.1 (111.4–194.1) | **0.005** |
| VFA reduction rate (%) | 51.2 (36.2–61.3) | 58.3 (39.2–65.9) | 42.6 (34.8–57.5) | 0.097 |
| Systolic blood pressure (mmHg) | 127.0 (117.8–140.0) | 127.0 (116.5–140.0) | 127.0 (119.0–136.0) | 0.934 |
| Diastolic blood pressure (mmHg) | 79.5 (69.8–86.3) | 80.0 (70.5–89.0) | 78.0 (69.0–85.0) | 0.479 |
| PRA (ng/mL/hr) | 1.0 (0.4–2.4) | 0.6 (0.4–1.4) | 1.5 (0.6–3.2) | 0.062 |
| PAC (pg/mL) | 97.0 (69.0–126.0) | 101.0 (70.5–117.5) | 92.5 (66.0–134.0) | 0.695 |
| Creatinine (mg/dL) | 0.8 (0.6–0.9) | 0.7 (0.6–0.8) | 0.8 (0.7–0.9) | **0.023** |
| HbA1c (%) | 5.5 (5.1–6.2) | 5.4 (5.1–5.7) | 5.9 (5.2–6.4) | 0.050 |
| HOMA-IR (no unit) | 1.4 (0.9–2.3) | 1.4 (0.7–1.8) | 1.4 (1.1–2.5) | 0.267 |
| HOMA-β (no unit) | 87.2 (50.7–140.9) | 89.2 (62.6–128.5) | 87.2 (42.2–144.0) | 0.455 |
| HDL-C (mg/dL) | 58.0 (48.0–66.0) | 62.0 (52.0–65.5) | 55.0 (43.5–66.3) | 0.147 |
| LDL-C (mg/dL) | 107.0 (91.0–128.0) | 117.0 (93.5–136.5) | 105.5 (89.8–114.5) | 0.313 |
| Uric acid (mg/dL) | 6.1 (5.0–7.3) | 6.3 (5.0–7.5) | 6.1 (5.2–7.1) | 0.974 |
| AHI (events/h) | 20.6 (9.5–39.6) | 12.9 (5.6–20.1) | 32.4 (17.4–48.6) | **<0.001** |

Values are median (interquartile range).

LSG, laparoscopic sleeve gastrectomy; SFA, subcutaneous fat area; VFA, visceral fat area; PRA, plasma renin activity; PAC, plasma aldosterone concentration; HbA1c, hemoglobin A1c; HOMA-IR, homeostasis model assessment of insulin resistance; HOMA-β; homeostasis model assessment of beta cell function; HDL-C, high-density lipoprotein cholesterol; LDL-C, low-density lipoprotein cholesterol; AHI, apnea hypopnea index; EWL, excess weight loss; TWL, total weight loss.

P values < 0.05 are shown in bold.

### Univariate and multivariate analyses correlated with HTN recurrence

In the univariate analysis, HbA1c was statistically significantly correlated with HTN recurrence. The numbers of preoperative ADs and VFA were correlated with HTN recurrence. Multivariate analysis revealed the absence of independent predictive factors for HTN recurrence.

### Stratification to predict the remission and recurrence of HTN

Based on the above results, patients were stratified into three sub-cohorts, according to the number of preoperative ADs and VFA (Table 5). Sub-cohort 1 had a high remission rate (100%) and low recurrence rate (16.7%). In contrast, sub-cohort 3 had poor remission (5.3%) and recurrence rates (100%).

## Discussion

In this study, we demonstrated the therapeutic effects of LSG in Japanese patients with severe obesity complicated by HTN. Stratification, combining the number of preoperative ADs and VFA, may be helpful for predicting HTN remission and recurrence post-LSG.

**Table 2. Univariate analysis using logistic regression models of preoperative factors associated with HTN remission.**

| Variable | Odds ratio | 95% CI | P value |
|---|---|---|---|
| Age (years) | 0.955 | 0.913–0.999 | **0.045** |
| Gender | 2.180 | 0.780–6.100 | 0.137 |
| Body weight (kg) | 0.971 | 0.944–0.998 | **0.039** |
| Body mass index (kg/m$^2$) | 0.888 | 0.772–1.020 | 0.095 |
| Obesity-related diseases | | | |
| Type 2 diabetes | 0.724 | 0.247–2.120 | 0.557 |
| Obstructive sleep apnea | — | — | — |
| Dyslipidemia | 0.918 | 0.296–2.850 | 0.883 |
| Hyperuricemia | 0.786 | 0.285–2.170 | 0.641 |
| Antihypertensive drug | 0.108 | 0.037–0.317 | **<0.001** |
| Duration of HTN (years) | 0.872 | 0.796–0.957 | **0.004** |
| SFA (cm$^2$) | 1.000 | 0.998–1.010 | 0.379 |
| VFA (cm$^2$) | 0.989 | 0.981–0.997 | **0.009** |
| Systolic blood pressure (mmHg) | 1.030 | 0.998–1.050 | 0.069 |
| Diastolic blood pressure (mmHg) | 1.040 | 1.000–1.090 | **0.031** |
| PRA (ng/mL/hr) | 0.966 | 0.900–1.040 | 0.339 |
| PAC (pg/mL) | 1.010 | 0.997–1.010 | 0.199 |
| Creatinine (mg/dL) | 0.237 | 0.049–1.150 | 0.074 |
| HbA1c (%) | 0.928 | 0.674–1.280 | 0.647 |
| HOMA-IR (no unit) | 1.060 | 0.962–1.170 | 0.238 |
| HOMA-β (no unit) | 1.000 | 0.997–1.010 | 0.515 |
| HDL-C (mg/dL) | 0.998 | 0.949–1.050 | 0.937 |
| LDL-C (mg/dL) | 1.010 | 0.997–1.030 | 0.113 |
| Uric acid (mg/dL) | 1.020 | 0.742–1.410 | 0.888 |
| AHI (events/h) | 0.970 | 0.949–0.993 | **0.009** |

HTN, hypertension; CI, confidence interval; SFA, subcutaneous fat area; VFA, visceral fat area; PRA, plasma renin activity; PAC, plasma aldosterone concentration; HbA1c, hemoglobin A1c; HOMA-IR, homeostasis model assessment of insulin resistance; HOMA-β; homeostasis model assessment of beta cell function; HDL-C, high-density lipoprotein cholesterol; LDL-C, low-density lipoprotein cholesterol; AHI, apnea hypopnea index.

P values < 0.05 are shown in bold.

The post-metabolic surgery remission rate of HTN is approximately 30–70% [3,4,6]. In the present study, the 12-month post-LSG HTN remission rate was approximately 40%, which is consistent with previous studies.

Factors, including age, the number of preoperative ADs, the duration of HTN, and excess weight loss, are associated with HTN remission [3,5,13–15]. Consistent with the findings of previous studies, patients in the remission cohort were significantly younger than those in the

**Table 3. Multivariate analysis using stepwise method of preoperative factors associated with HTN remission.**

| Variable | Odds ratio | 95% CI | P value |
|---|---|---|---|
| Antihypertensive drug | 0.172 | 0.059–0.505 | **0.001** |
| VFA (cm$^2$) | 0.986 | 0.972–0.999 | **0.040** |

HTN, hypertension; CI, confidence interval; VFA, visceral fat area.

P values < 0.05 are shown in bold.

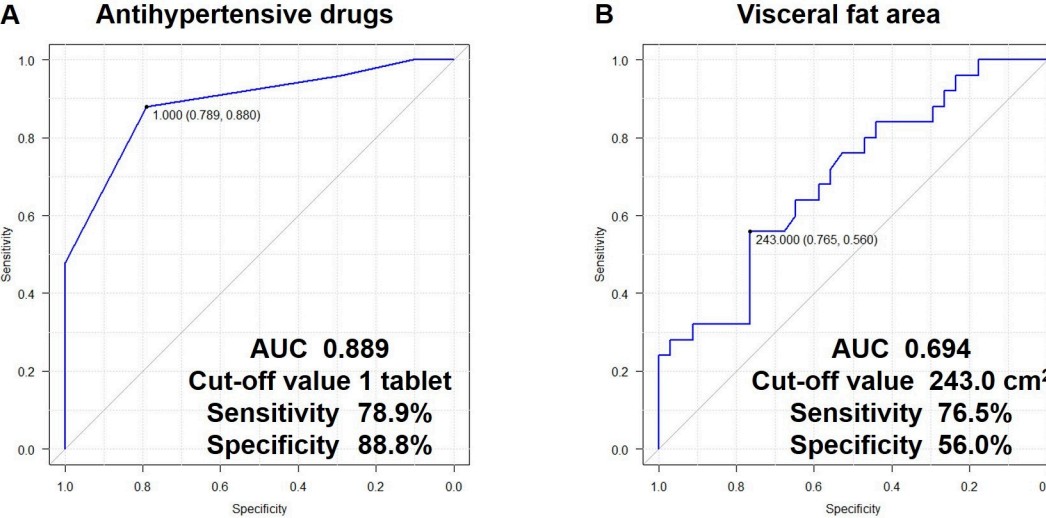

**Fig 2.** Receiver operating characteristic curve analysis of preoperative antihypertensive drugs (A) and visceral fat area (B). Preoperative antihypertensive drugs can be a good predictor of hypertension remission after laparoscopic sleeve gastrectomy. AUC, area under the curve.

non-remission cohort. Moreover, the number of preoperative ADs was smaller, the duration of HTN was shorter, and the postoperative effects of weight loss improved in the remission cohort compared to in the non-remission cohort.

In addition, multivariate analyses revealed that the number of preoperative ADs and VFA were independent predictors of HTN remission. Visceral fat plays an essential role in the pathophysiology of HTN; however, few studies have examined visceral fat and the remission of HTN post-metabolic surgery [16,17]. Auclair et al. found that patients with severe obesity, with HTN remission, after biliopancreatic diversion with duodenal switch surgery have better visceral fat reduction than patients who were not in HTN remission [16]. In this study, the VFA was significantly smaller in the HTN remission cohort than in the non-remission cohort at baseline and 6 and 12 months post-LSG. The findings of several previous studies and those of the present study indicate the importance of measuring VFA in predicting the effect of metabolic surgery on improving HTN. As mentioned above, the VFA reduction rate is also an important indicator. In the present study, the VFA reduction rate at 6 months post-LSG was significantly higher in the HTN remission cohort than in the non-remission cohort. The rate of VFA reduction appeared to be more reasonable for predicting HTN remission than the absolute VFA value. However, the concept of this study was to extract factors that predict HTN remission after LSG from preoperative clinical parameters, and the absolute value of preoperative VFA was used as the evaluation value instead of the VFA reduction rate.

A study that evaluated HTN remission after Roux-en-Y gastric bypass (RYGB) reported that HTN duration was an independent predictor of HTN remission. For each additional year of HTN duration, the chance of HTN remission decreased by approximately 15% after RYGB [15]. In this study, HTN duration was not identified as an independent predictor of HTN remission, but the duration of HTN was significantly shorter in the HTN remission cohort than in the non-remission cohort. Undoubtedly, the duration of HTN is also an important factor associated with HTN remission.

Few reports on the recurrence of HTN post-metabolic surgery exist [3,5]. The post-metabolic surgery recurrence rate of HTN is approximately 20–60%, which is consistent with the

**Table 4. Comparison of patient characteristics and clinical parameters at baseline, 12 months, and 36 months after LSG between the recurrence and the non-recurrence groups.**

| Variable | All patients (n = 25) | Recurrence group (n = 9) | Non-recurrence group (n = 16) | P value |
|---|---|---|---|---|
| Baseline | | | | |
| Age (years) | 41.0 (33.0–51.0) | 38.0 (33.0–51.0) | 41.0 (31.8–49.8) | 0.821 |
| Gender, n (%) | | | | |
| Male | 10 (41.7) | 3 (42.9) | 7 (41.2) | |
| Female | 14 (58.3) | 4 (57.1) | 10 (58.8) | 1.000 |
| Body weight (kg) | 107.0 (93.0–129.0) | 107.0 (94.0–134.0) | 106.0 (93.0–125.3) | 0.495 |
| Body mass index (kg/m$^2$) | 37.0 (35.0–40.0) | 38.0 (36.0–40.0) | 36.5 (35.0–39.3) | 0.362 |
| Obesity-related diseases, n (%) | | | | |
| Type 2 diabetes | 16 (64.0) | 5 (55.6) | 11 (68.8) | 0.671 |
| Obstructive sleep apnea | 25 (100) | 9 (100) | 16 (100) | — |
| Dyslipidemia | 18 (72.0) | 6 (66.7) | 12 (75.0) | 0.673 |
| Hyperuricemia | 11 (44.0) | 5 (55.6) | 6 (37.5) | 0.434 |
| Antihypertensive drug | 1 (0–1) | 1 (0–2) | 0 (0–1) | 0.099 |
| Duration of HTN (years) | 1.0 (1.0–11.0) | 1.0 (1.0–12.0) | 1.0 (1.0–6.5) | 0.864 |
| SFA (cm$^2$) | 525.0 (399.0–621.0) | 527.0 (399.0–584.0) | 521.5 (403.5–644.5) | 0.955 |
| VFA (cm$^2$) | 231.0 (196.0–290.0) | 290.0 (226.0–323.0) | 222.0 (167.3–258.8) | 0.074 |
| Systolic blood pressure (mmHg) | 146.0 (138.0–157.0) | 140.0 (137.0–152.0) | 149.0 (139.5–158.3) | 0.444 |
| Diastolic blood pressure (mmHg) | 91.0 (83.0–95.0) | 85.0 (82.0–90.0) | 92.5 (88.3–95.8) | 0.360 |
| PRA (ng/mL/hr) | 1.2 (0.9–2.9) | 1.0 (0.5–1.8) | 1.7 (1.0–4.2) | 0.142 |
| PAC (pg/mL) | 117.0 (105.8–224.3) | 156.0 (86.0–269.0) | 115.0 (112.0–211.0) | 0.937 |
| Creatinine (mg/dL) | 0.7 (0.6–0.8) | 0.7 (0.7–0.8) | 0.7 (0.6–0.8) | 0.335 |
| HbA1c (%) | 6.7 (5.8–8.1) | 5.8 (5.5–6.7) | 7.5 (6.1–9.0) | **0.015** |
| HOMA-IR (no unit) | 6.0 (3.5–7.5) | 4.0 (3.5–5.5) | 6.5 (3.8–9.3) | 0.253 |
| HOMA-β (no unit) | 127.0 (87.0–235.0) | 194.0 (122.0–256.5) | 114.5 (69.5–203.8) | 0.175 |
| HDL-C (mg/dL) | 43.0 (36.0–49.0) | 43.0 (42.0–47.0) | 41.0 (34.8–49.5) | 0.514 |
| LDL-C (mg/dL) | 112.0 (104.0–152.0) | 135.0 (112.0–167.0) | 109.0 (102.0–133.8) | 0.100 |
| Uric acid (mg/dL) | 6.7 (5.6–7.7) | 6.7 (6.2–7.7) | 7.0 (5.6–7.7) | 0.843 |
| AHI (events/h) | 38.5 (25.0–61.5) | 38.0 (25.0–62.5) | 38.5 (18.8–59.3) | 0.811 |
| 12 months after LSG | | | | |
| Body weight (kg) | 72.0 (68.0–88.0) | 73.0 (68.0–88.0) | 71.0 (68.5–86.0) | 0.654 |
| Body mass index (kg/m2) | 28.5 (26.0–31.0) | 29.0 (27.0–32.0) | 28.0 (26.0–30.5) | 0.590 |
| EWL (%) | 62.5 (53.8–73.0) | 62.0 (52.0–73.0) | 63.0 (56.5–72.5) | 0.676 |
| TWL (%) | 29.5 (24.8–32.3) | 28.0 (26.0–31.0) | 30.0 (24.5–34.0) | 0.654 |
| SFA (cm$^2$) | 313.9 (166.8–396.2) | 313.9 (209.1–413.2) | 286.1 (155.0–364.0) | 0.600 |
| VFA (cm$^2$) | 104.3 (69.8–145.3) | 122.1 (104.2–142.1) | 91.5 (60.5–167.3) | 0.403 |
| VFA reduction rate (%) | 58.3 (39.2–65.9) | 56.7 (38.5–69.0) | 58.3 (41.6–59.8) | 0.975 |
| Systolic blood pressure (mmHg) | 127.0 (116.5–140.0) | 140.0 (140.0–143.0) | 121.5 (113.5–126.8) | **0.002** |
| Diastolic blood pressure (mmHg) | 80.0 (70.5–89.0) | 86.0 (80.0–89.0) | 71.0 (69.0–85.0) | 0.063 |
| PRA (ng/mL/hr) | 0.6 (0.4–1.4) | 0.4 (0.3–0.6) | 1.0 (0.5–3.0) | **0.025** |
| PAC (pg/mL) | 101.0 (70.5–117.5) | 89.0 (71.5–119.3) | 101.0 (72.0–109.0) | 0.860 |
| Creatinine (mg/dL) | 0.7 (0.6–0.8) | 0.7 (0.6–0.8) | 0.7 (0.6–0.8) | 0.511 |
| HbA1c (%) | 5.4 (5.1–5.7) | 5.4 (5.0–5.4) | 5.5 (5.2–5.8) | 0.230 |
| HOMA-IR (no unit) | 1.4 (0.7–1.8) | 1.4 (1.2–1.8) | 1.3 (0.6–2.0) | 0.785 |
| HOMA-β (no unit) | 89.2 (62.6–128.5) | 118.2 (70.1–136.6) | 75.7 (62.6–119.3) | 0.616 |
| HDL-C (mg/dL) | 62.0 (52.0–65.5) | 57.0 (50.0–64.0) | 62.5 (55.8–69.0) | 0.360 |
| LDL-C (mg/dL) | 117.0 (93.5–136.5) | 117.0 (96.0–141.0) | 115.0 (92.8–132.0) | 0.682 |

*(Continued)*

**Table 4.** (Continued)

| Variable | All patients (n = 25) | Recurrence group (n = 9) | Non-recurrence group (n = 16) | P value |
|---|---|---|---|---|
| Uric acid (mg/dL) | 6.3 (5.0–7.5) | 5.0 (4.3–5.7) | 6.8 (6.1–7.7) | 0.068 |
| AHI (events/h) | 12.9 (5.6–20.1) | 13.9 (10.9–20.1) | 9.2 (2.9–15.0) | 0.407 |
| 36 months after LSG | | | | |
| Body weight (kg) | 83.0 (74.5–91.5) | 82.5 (76.5–95.8) | 83.0 (73.0–90.5) | 0.772 |
| Body mass index (kg/m2) | 32.0 (28.3–34.0) | 34.0 (28.5–35.5) | 32.0 (28.0–34.0) | 0.749 |
| EWL (%) | 56.0 (35.0–63.5) | 53.0 (41.8–65.8) | 56.0 (30.0–61.0) | 0.649 |
| TWL (%) | 24.0 (19.0–28.5) | 26.0 (21.8–28.8) | 23.0 (15.5–28.5) | 0.619 |
| Recurrent weight gain | 4 (16.0) | 1 (11.1) | 3 (18.8) | 1.000 |
| SFA (cm$^2$) | 328.7 (205.9–553.5) | 456.9 (316.3–648.8) | 264.5 (189.7–425.8) | 0.429 |
| VFA (cm$^2$) | 124.8 (94.9–157.2) | 142.7 (125.6–171.6) | 94.9 (88.4–103.5) | 0.052 |
| VFA reduction rate (%) | 55.8 (32.5–59.9) | 55.8 (24.1–56.7) | 47.3 (37.4–61.6) | 0.537 |
| Systolic blood pressure (mmHg) | 124.5 (113.8–134.0) | 132.0 (121.3–136.5) | 121.5 (112.5–126.3) | 0.172 |
| Diastolic blood pressure (mmHg) | 80.0 (71.8–85.3) | 81.5 (78.3–90.5) | 76.0 (68.0–85.0) | 0.292 |
| PRA (ng/mL/hr) | 0.6 (0.5–1.1) | 0.5 (0.3–0.5) | 0.7 (0.6–2.8) | **0.043** |
| PAC (pg/mL) | 122.5 (88.5–146.5) | 145.0 (111.5–175.3) | 102.5 (76.5–129.8) | 0.343 |
| Creatinine (mg/dL) | 0.7 (0.7–0.7) | 0.7 (0.7–0.7) | 0.7 (0.6–0.8) | 0.643 |
| HbA1c (%) | 5.5 (5.2–5.8) | 5.4 (5.2–5.5) | 5.9 (5.2–6.6) | 0.268 |
| HOMA-IR (no unit) | 1.2 (1.0–1.8) | 1.3 (1.0–1.6) | 1.2 (0.8–4.2) | 1.000 |
| HOMA-β (no unit) | 91.5 (49.3–129.0) | 116.4 (63.7–135.0) | 77.5 (43.5–105.6) | 0.310 |
| HDL-C (mg/dL) | 59.0 (53.8–66.5) | 57.0 (48.0–69.3) | 59.0 (56.5–62.3) | 0.598 |
| LDL-C (mg/dL) | 133.5 (104.0–141.3) | 135.5 (123.5–140.3) | 130.5 (98.3–145.3) | 0.878 |
| Uric acid (mg/dL) | 5.6 (5.0–6.9) | 5.3 (4.5–6.3) | 5.9 (5.2–6.9) | 0.460 |
| AHI (events/h) | 8.5 (5.6–15.9) | 14.2 (9.5–17.0) | 6.0 (4.0–8.0) | 0.252 |

Values are median (interquartile range).

LSG, laparoscopic sleeve gastrectomy; SFA, subcutaneous fat area; VFA, visceral fat area; PRA, plasma renin activity; PAC, plasma aldosterone concentration; HbA1c, hemoglobin A1c; HOMA-IR, homeostasis model assessment of insulin resistance; HOMA-β; homeostasis model assessment of beta cell function; HDL-C, high-density lipoprotein cholesterol; LDL-C, low-density lipoprotein cholesterol; AHI, apnea hypopnea index; EWL, excess weight loss; TWL, total weight loss.

P values < 0.05 are shown in bold.

findings of this study. As with HTN remission, factors, including the number of preoperative ADs and the duration of HTN are correlated with HTN recurrence [3,5].

In this study, independent predictors of HTN recurrence could not be identified; however, in the univariate analyses, the number of preoperative ADs and VFA were correlated with HTN recurrence. Interestingly, our study revealed that HTN complicated by severe obesity was more likely to achieve remission and less likely to recur post-LSG in patients with a low number of preoperative ADs and VFA. These results should be validated by future studies,

**Table 5. Stratification to predict remission and recurrence of HTN after LSG.**

| | HTN remission rate within 12 months | HTN recurrence rate within 36 months | Definition |
|---|---|---|---|
| Group 1 | 100% | 16.7% | Antihypertensive drugs ≤ 1 tablet and VFA ≤ 243 cm$^2$ |
| Group 2 | 44.4% | 50.0% | Other than Group1 and Group 3 |
| Group 3 | 5.3% | 100% | Antihypertensive drugs > 1 tablet and VFA > 243 cm$^2$ |

HTN, hypertension; LSG, laparoscopic sleeve gastrectomy; VFA, visceral fat area.

using larger sample sizes. The preoperative HbA1c level was statistically significantly lower in the recurrence cohort than in the non-recurrence cohort; however, this may have been due to the small sample size.

Several biological mechanisms have been proposed to explain the correlation between visceral fat and HTN. First, adipokines secreted by adipocytes, such as leptin, play an essential role in the progression of HTN. Leptin is known to regulate appetite and energy expenditure and has been suggested to act directly on the sympathetic nervous system to increase blood pressure [18]. Bełtowski et al. observed that leptin-induced natriuresis is impaired in obese rats [19]. This finding suggests that sodium retention is associated with HTN in patients with obesity.

Second, an increase in perirenal fat, one of the components of abdominal visceral fat, reduces blood flow in the vasa recta and loop of Henle, by compressing the kidneys. This increases renin secretion and sodium reabsorption in the proximal tubules [18]. In addition, leptin secreted by perirenal adipocytes may be associated with the development of HTN. A study by Li et al. found that a long-term reduction in blood pressure is achieved by bilateral ablation or denervation of the perirenal adipose tissue in rats with spontaneous HTN [20].

Third, an increase in visceral fat correlated with obesity affects the renin–angiotensin–aldosterone system (RAAS) [21]. Angiotensinogen, a precursor of angiotensin I, is mainly produced in the liver; visceral fat is recognized as a major source of extrahepatic angiotensinogen [22]. Rahmouni et al. observed a significant increase in angiotensinogen gene expression in the intra-abdominal fat of obese mice fed a high-fat diet [21]. In addition, a study by Yiannikouris et al. revealed that an adipocyte-specific deficiency in angiotensinogen resulted in a decrease in the concentration of plasma angiotensinogen and systolic blood pressure in mice [22]. Thus, theoretically, an increase in visceral fat with obesity directly activates the RAAS and affects the pathophysiology of HTN. In support of this, even in this study, the improvement in post-LSG HTN was greater in patients with low VFA levels.

Moreover, RAAS is critical factor directly associated with HTN. In Conn's syndrome, which is caused by an adrenal adenoma autonomously producing aldosterone, severe treatment-resistant HTN is manifested [23]. RAAS is a significant target for treating of HTN, and related ADs are routinely used. Additionally, RAAS affects the insulin signaling pathway and is involved in glucose homeostasis [24]. Although not shown in the results section, preoperative PAC also tended to be negatively correlated with HbA1c and positively correlated with HOMA-β in this study (S1 Table). As described above, the pathophysiology of HTN in patients with severe obesity involves a complex interaction of multiple biological mechanisms.

In the present study, the PRA levels were significantly lower at 12 and 36 months post-LSG in the HTN recurrence cohort than in the non-recurrence cohort. PRA is generally reduced in patients with HTN, caused by an excess body sodium content [25]. Furthermore, importantly, some patients in the HTN recurrence cohort took angiotensin II receptor blockers orally post-recurrence. Additionally, salt intake is correlated with PRA levels [26]; however, we could not analyze this correlation because it was outside the scope of this study.

We have previously reported that serum transforming growth factor-beta (TGF-β)-related proteins fluctuated dynamically pre- and post-LSG [27]. These include proteins such as EMILIN-1, which have been implicated in the pathogenesis of HTN [28]. Findings of a previous study revealed an increase in systemic blood pressure, with narrowing of the peripheral arteries in EMILIN-1-deficient mice [28]. Thus, various biological mechanisms may complicate the pathophysiology of HTN in obesity, and further research is needed to elucidate the pathophysiology.

Whether differences in the improvement of HTN exist between different surgical procedures remains controversial. In a prospective cohort study involving 787 patients, comparing

the effects of sleeve gastrectomy and one-anastomosis gastric bypass in improving HTN, no differences were observed in HTN remission or recurrence between the two surgical methods [29]. Conversely, in a 10-year follow-up study comparing the effects of LSG and laparoscopic Roux-en-Y gastric bypass (LRYGB) in improving postoperative HTN, the proportion of patients who discontinued medication was higher in the group of patients who underwent LRYGB [30]. In another 10-year large observational study in Sweden, sleeve gastrectomy was associated with a higher risk of HTN recurrence than RYGB [31]. For long-term treatment effects on HTN, bypass surgery may be superior to sleeve gastrectomy; nonetheless, further research is required.

This study has some limitations. First, this was a retrospective study conducted at a single institution; thus, the small sample size cannot be disregarded. Second, in the present study, the patient's blood pressure was measured in a hospital. White-coat HTN may cause blood pressure to be higher than usual. Ideally, blood pressure should be measured at home. Third, CT images were used to measure the VFA in this study; thus, CT included the risk of radiation exposure and the need for image analysis software. A more straightforward indicator than the VFA, calculated by analyzing CT images, is required. Fourth, this study was conducted on patients with severe obesity in Japan, and the results do not necessarily extrapolate to patients in other countries.

In conclusion, we identified the factors correlated with HTN remission and recurrence post-LSG in patients with severe obesity. By combining the number of preoperative ADs and VFA, stratification may help predict HTN remission and recurrence post-LSG.

## Supporting information

**S1 Table. Relationship between plasma aldosterone concentration and type 2 diabetes-related parameters.**
(DOCX)

## Author Contributions

**Conceptualization:** Hideki Kumagai, Akira Sasaki, Akira Umemura, Taro Ando, Hirokatsu Katagiri, Hiroyuki Nitta.

**Data curation:** Hideki Kumagai, Yota Tanahashi, Takafumi Iwasaki.

**Funding acquisition:** Akira Sasaki, Akira Umemura, Taro Ando.

**Methodology:** Akira Umemura, Hirokatsu Katagiri, Hiroyuki Nitta.

**Project administration:** Akira Sasaki.

**Supervision:** Akira Sasaki.

**Visualization:** Hideki Kumagai, Yota Tanahashi, Takafumi Iwasaki.

**Writing – original draft:** Hideki Kumagai.

**Writing – review & editing:** Akira Sasaki, Akira Umemura.

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
