## [Decision Letter · Decision Letter 0]

9 Sep 2024

PONE-D-24-29682Predictive Factors Involving the Remission and Recurrence of Hypertension Post-Laparoscopic Sleeve Gastrectomy in Japanese Patients with Severe ObesityPLOS ONE

Dear Dr. Kumagai,

Thank you for submitting your manuscript to PLOS ONE. After careful consideration, we feel that it has merit but does not fully meet PLOS ONE’s publication criteria as it currently stands. Therefore, we invite you to submit a revised version of the manuscript that addresses the points raised during the review.

We look forward to receiving your revised manuscript.

Kind regards,

Tatsuo Shimosawa, M.D., Ph.D.

Academic Editor

PLOS ONE

Journal Requirements:

"This work was supported by the following grants: JSPS KAKENHI Grant Numbers 22K11812, 23K10852; Research Grant from Keiryokai Grant Number 148."

3. Please expand the acronym “JSPS” (as indicated in your financial disclosure) so that it states the name of your funders in full.

"The authors declare that there is no conflict of interest regarding the publication of this article."

5. In this instance it seems there may be acceptable restrictions in place that prevent the public sharing of your minimal data. However, in line with our goal of ensuring long-term data availability to all interested researchers, PLOS’ Data Policy states that authors cannot be the sole named individuals responsible for ensuring data access (http://journals.plos.org/plosone/s/data-availability#loc-acceptable-data-sharing-methods).

Additional Editor Comments:

Three experts and a statistician have concerns on your article.

Reviewers' comments:

Reviewer's Responses to Questions

**Comments to the Author**

1. Is the manuscript technically sound, and do the data support the conclusions?

Reviewer #1: Yes

Reviewer #2: No

Reviewer #3: Partly

Reviewer #4: Yes

2. Has the statistical analysis been performed appropriately and rigorously? 

Reviewer #1: Yes

Reviewer #2: No

Reviewer #3: I Don't Know

Reviewer #4: Yes

3. Have the authors made all data underlying the findings in their manuscript fully available?

Reviewer #1: Yes

Reviewer #2: Yes

Reviewer #3: Yes

Reviewer #4: Yes

4. Is the manuscript presented in an intelligible fashion and written in standard English?

Reviewer #1: Yes

Reviewer #2: Yes

Reviewer #3: No

Reviewer #4: Yes

5. Review Comments to the Author

Reviewer #1: This paper is highly significant as it examines the resolution of hypertension and the contributing factors in a large cohort of Japanese patients with severe obesity who underwent bariatric surgery. The study investigates numerous factors, including endocrinological aspects, and employs a rigorous research methodology.

The following points should be considered:

1. Has the study ensured that the patient group consists only of patients with essential hypertension, excluding those with secondary hypertension, including adrenal-related hypertension?

2. The degree of visceral obesity and comorbidities prior to surgery plays a crucial role in the remission of hypertension. How does the reduction in visceral fat correlate with the remission rate?

3. Visceral fat is suspected to be associated with chronic inflammation. What is the relationship between changes in high-sensitivity CRP and this condition?

4. It has been reported that preoperative blood aldosterone levels in severely obese patients are related to insulin resistance and subsequent improvements in glucose tolerance. How does this relate to hypertension?

Reviewer #2: The authors concluded that " In Japanese patients with severe obesity, the number of preoperative ADs and VFA

were correlated with HTN remission and recurrence post-LSG. Stratification, by combining the number of preoperative ADs and VFA can be useful in predicting HTN remission and recurrence."

The numbers of patients are too small and there are some problem in Statistical Analyses (for example variables are too many for patients).

Reviewer #3: DEAR Authors, thanks for your submission. I have some comments and questions for you.

- You should ask for a professional revision to correct grammatical errors in your manuscript. Ex.: "Metabolic surgery, inclusive of laparoscopic sleeve gastrectomy (LSG) can be used to treat hypertension (HTN); however, insufficient deliberation exists regarding the therapeutic effect of post-metabolic surgery on HTN."

- About this phrase, You cannot say that LSG can be used to treat HTN because BS is a adjuvant therapy to treat patients with hypertension linked to obesity.

- You must not use "patients who are severely obese". Change to patients with severe obesity. Check all the manuscript.

- definition of weight regain should follow the IFSO definitions for publication: https://media.springer.com/full/springer-instructions-for-authors-assets/pdf/11695_IFSO%20DEFINITIONS_May_2024.pdf

- You stated that your study is retrospective. I would like to know if this exams: CT for the subcutaneous fat area (SFA) and visceral fat area (VFA), plasma renin activity (PRA), plasma aldosterone concentration (PAC) are in your routine. It seems to me that these are not usual exams for a bariatric patient. Can you explain?

- "Patient Characteristics Correlated with HTN Remission" and "Comparative Analysis of the Clinical Parameters between the Remission and Non-Remission Cohorts" should be together. there is no reason to separate all these parameters. And, it is better to make one table for each period with all parameters. Same observation for recurrence.

- About the VFA, we know that it is important factor for the HTN related to obesity. My suggestion to you is that you should include the reduction in VFA from the baseline instead the absolute value because it is more important the amount of VFA reduction than the absolute level at each time of follow-up. And, it seems to me that your finding is the opposite that I would expect because patients with less VFA tend to have other causes for their hypertension that could not be solved by loosing weight. Can you discuss this issue?

- bout the limitations of your study: the number of patients is really a big issue. I am afraid that with a few patients it is hard to make all your conclusions. Second: "Second, in the present study, the patient’s blood pressure was measured in a hospital." I am in doubt if you make the diagnosis of hypertension in the hospital before the surgery. Did this occur?

- Replace the reference abouT the GATEWAY TRIAL (4) by the last publication: J A C C V OL . 8 3 , F E B R U A R Y 1 3 , 2 0 2 4 : 6 3 7 – 6 4 8

- Include 2 important references and discuss: PLoS Med 18(11): e1003817. https://doi.org/10.1371/journal.pmed.1003817 and Obes Surg 2023 Aug;33(8):2485-2492. doi: 10.1007/s11695-023-06711-2

- the PLOS Med paper stated that Sleeve gastrectomy is a factor for recurrence.

Reviewer #4: The authors aimed to identify risk factors and build the prediction for HTN remission and recurrence post-LSG in severely obese patients with 62 patients. The results showed that the number of preoperative antihypertensive drugs and visceral fat area were associated and predictive of HTN remission and recurrence post-LSG.

1. Line 84. “diagnosed with HTN preoperatively and were taking oral…..” The definition of HTN should be “or”?

2. Sample recreation criteria. Is there any age restriction? Were pregnant women excluded? Please provide exclusion and inclusion criteria in the manuscript.

3. Why did the authors decide to present data as medians (interquartile ranges) instead of mean(SD)?

4. Why was the Mann-Whitney U test used for the continuous variable? Also, the Mann-Whitney U test is a non-parametric test and is not intended for comparing the mean value.

6. PLOS authors have the option to publish the peer review history of their article (what does this mean?). If published, this will include your full peer review and any attached files.

Reviewer #1: No

Reviewer #2: No

Reviewer #3: No

Reviewer #4: No

---

## [Author Response · Author response to Decision Letter 0]

7 Oct 2024

Prof. Emily Chenette

PLOS ONE, Editor in Chief

Dear Prof. Chenette,

The comments of the four reviewers have been very helpful for us to revise our manuscript. We have attempted to address the questions raised by the reviewers. Our point-by-point responses to the reviewers’ comments are provided below.

Reviewer #1: This paper is highly significant as it examines the resolution of hypertension and the contributing factors in a large cohort of Japanese patients with severe obesity who underwent bariatric surgery. The study investigates numerous factors, including endocrinological aspects, and employs a rigorous research methodology.

The following points should be considered:

1. Has the study ensured that the patient group consists only of patients with essential hypertension, excluding those with secondary hypertension, including adrenal-related hypertension?

Reply: 

Thank you for your valuable indication. This study examined essential hypertension, and excluded secondary hypertension. We added it in the section of Methods (Page 4, Line 85-87).

2. The degree of visceral obesity and comorbidities prior to surgery plays a crucial role in the remission of hypertension. How does the reduction in visceral fat correlate with the remission rate?

Reply：

We examined the relationship between the VFA reduction rate and the HTN remission rate, and added it to the manuscript (Page 7, Line 135-136) and table (Table 1, 4). As you pointed out, VFA reduction rate tended to be higher in the HTN remission cohort than in the non-remission cohort. 

3. Visceral fat is suspected to be associated with chronic inflammation. What is the relationship between changes in high-sensitivity CRP and this condition?

Reply：

We examined the relationship between VFA and high-sensitivity CRP. The Spearman correlation coefficients for VFA and high-sensitivity CRP at baseline and 6 and 12 months post-LSG were 0.131, 0.146, and 0.269, respectively, and there was no significant association in our cohort. However, as you mentioned, visceral fat releases inflammatory cytokines, which have been reported to be associated with many obesity-related diseases. In cohorts with large sample sizes, we think that it is possible to clarify an association between the two. 

4. It has been reported that preoperative blood aldosterone levels in severely obese patients are related to insulin resistance and subsequent improvements in glucose tolerance. How does this relate to hypertension?

Reply：

You raised an important suggestion, and we examined plasma aldosterone concentration (PAC) and type 2 diabetes related parameters. PAC and HbA1c tended to be negatively correlated (Spearman correlation coefficient: -0.4~-0.2), and PAC and HOMA-β tended to be positively correlated (Spearman correlation coefficient: 0.2~0.35), but the relationship with HOMA-IR, which is an indicator of insulin resistance, was not clear. 

On the other hand, the question of whether preoperative PAC can predict HTN remission within 12 months postoperatively was found to be area under the curve 0.488 (95% confidence interval 0.306 - 0.670) in receiver operating characteristic analysis, indicating that preoperative PAC is not a useful predictor of HTN remission. 

We added the important physiological associations that you raised and above results to the discussion section (Page 21, Line 277-282 and S1 Table). The association between PAC and diabetes/hypertension remission was less clear in this cohort. However, as mentioned in the discussion, visceral fat is involved in insulin resistance and is a major source of extrahepatic angiotensinogen; thus, visceral fat is considered to play an important role in the pathophysiology of hypertension through glucose metabolism and the renin–angiotensin–aldosterone system.

Reviewer #2: The authors concluded that " In Japanese patients with severe obesity, the number of preoperative ADs and VFA were correlated with HTN remission and recurrence post-LSG. Stratification, by combining the number of preoperative ADs and VFA can be useful in predicting HTN remission and recurrence."

The numbers of patients are too small and there are some problems in Statistical Analyses (for example variables are too many for patients).

Reply：

Thank you for your very important suggestion related to statistical analysis. One of the limitations of this study is the small sample size. Therefore, we performed multivariate analysis with a stepwise method, and two variables, which was the number of preoperative antihypertensive drugs and VFA, remained. As you mentioned, the previous table was misunderstood as having too many variables, so we revised the table to include only the number of preoperative antihypertensive drugs and VFA (Table 3). Additionally, in response to the suggestion that the sample size was too small to conclude that the number of preoperative ADs and VFA in this were useful predictors of HTN remission and recurrence, the concluding remarks were changed as follows: By combining the number of preoperative ADs and VTA, stratification may help predict HTN remission and recurrence post-LSG (Page 22, Line 317-318). We would appreciate it if you could check it. 

Reviewer #3: DEAR Authors, thanks for your submission. I have some comments and questions for you.

- You should ask for a professional revision to correct grammatical errors in your manuscript. Ex.: "Metabolic surgery, inclusive of laparoscopic sleeve gastrectomy (LSG) can be used to treat hypertension (HTN); however, insufficient deliberation exists regarding the therapeutic effect of post-metabolic surgery on HTN."

- About this phrase, you cannot say that LSG can be used to treat HTN because BS is a adjuvant therapy to treat patients with hypertension linked to obesity.

Reply：

Thank you for your suggestion. The relevant phrase was corrected as follows: Metabolic surgery, including laparoscopic sleeve gastrectomy (LSG), not only has a weight loss effect but may also improve HTN complicated by severe obesity (Page 3, Line 52-53). 

- You must not use "patients who are severely obese". Change to patients with severe obesity. Check all the manuscript.

Reply：

Thank you for your important feedback. The manuscript has been corrected as you pointed out.

- definition of weight regain should follow the IFSO definitions for publication: https://media.springer.com/full/springer-instructions-for-authors-assets/pdf/11695_IFSO%20DEFINITIONS_May_2024.pdf

Reply：

Thank you for your valuable suggestion. In accordance with the IFSO definition, weight regain in this study was defined as weight gain of more than 30 % of initial surgical weight loss or worsening of an obesity-related disease that was a significant indication for surgery. Accordingly, the relevant text and table descriptions have been corrected (Page 5, Line 100-103 and Table 4).

- You stated that your study is retrospective. I would like to know if these exams: CT for the subcutaneous fat area (SFA) and visceral fat area (VFA), plasma renin activity (PRA), plasma aldosterone concentration (PAC) are in your routine. It seems to me that these are not usual exams for a bariatric patient. Can you explain?

Reply：

We have been conducting multiple studies on the effects of laparoscopic sleeve gastrectomy on weight loss and metabolic improvement (Sasaki A, et al. Biomedicines. 2022 Feb 15;10(2):453 and Umemura A, et al. Surg Today. 2020 Sep;50(9):1056-1064., etc.). In addition, most patients with severe obesity suffer from obstructive sleep apnea (Yanari S, et al. J Diabetes Investig. 2022 Jun;13(6):1073-1085.), and it is important to assess the risk of cardiovascular disease. Therefore, we obtain the consent of patient to the examination as soon as possible, perform the above examinations, and comprehensive treatment in cooperation with cardiologists and endocrinologists as necessary. 

- "Patient Characteristics Correlated with HTN Remission" and "Comparative Analysis of the Clinical Parameters between the Remission and Non-Remission Cohorts" should be together. there is no reason to separate all these parameters. And, it is better to make one table for each period with all parameters. Same observation for recurrence.

Reply：

As you mentioned, we have combined "Patient Characteristics Correlated with HTN Remission" and Comparative Analysis of the Clinical Parameters between the Remission and Non-Remission Cohorts" (Page 6, Line 128). The same applies to recurrence (Page 12, Line 171). We would appreciate it if you could check it.

- About the VFA, we know that it is important factor for the HTN related to obesity. My suggestion to you is that you should include the reduction in VFA from the baseline instead the absolute value because it is more important the amount of VFA reduction than the absolute level at each time of follow-up. And, it seems to me that your finding is the opposite that I would expect because patients with less VFA tend to have other causes for their hypertension that could not be solved by losing weight. Can you discuss this issue?

Reply：

As you mentioned, we have added the VFA reduction rate at each timing to the tables (Table 1, 4). As for whether the VFA reduction rate at 6 months postoperatively predicted HTN remission, the receiver operating characteristic (ROC) analysis showed an area under the curve (AUC) of 0.719 (95% confidence interval [CI]: 0.579-0.859). This result was slightly better compared to the AUC 0.694 (95% CI: 0.558-0.830) for the preoperative VFA. In addition, as for whether the VFA reduction rate at 6 months postoperatively predicted HTN recurrence, the AUC of the VFA reduction rate was 0.611 (95% CI: 0.371-0.852). This result was inferior compared to the AUC of 0.722 (95％ CI: 0.501-0.943) for the preoperative VFA.

The reason why the VFA reduction rate is inferior to the absolute value in recurrence is probably due to the sample size, and we agree that the VFA reduction rate is more useful than the absolute value, as you mentioned. However, the concept of this study was to extract factors that predict HTN remission and recurrence after LSG in the preoperative clinical parameters, and the absolute preoperative VFA was used for the stratification instead of the VFA reduction rate. We added the above to the Discussion section (Page 18-19, Line 223-230).

The reason for the higher postoperative HTN remission rate in those with lower absolute VFA levels was considered that the cohort with a large VFA had more comorbidities (mean: 3.0 versus 3.7, P = 0.006). Visceral fat affects insulin resistance and releases inflammatory cytokines that are associated with various obesity-related diseases; thus, we consider that patients with high levels of VFA are less likely to achieve HTN remission after metabolic surgery such as sleeve gastrectomy.

- bout the limitations of your study: the number of patients is really a big issue. I am afraid that with a few patients it is hard to make all your conclusions. 

Reply：

As you mentioned, the small sample size is a major limitation of this study. The wording was changed as follows: By combining the number of preoperative ADs and VTA, stratification may help predict HTN remission and recurrence post-LSG (Page 22, Line 317-318).

Second: "Second, in the present study, the patient’s blood pressure was measured in a hospital." I am in doubt if you make the diagnosis of hypertension in the hospital before the surgery. Did this occur?

Reply：

In our hospital, blood pressure is measured routinely when patients with severe obesity visiting a hospital. This is because, as mentioned above, most patients with severe obesity suffer from obstructive sleep apnea, and we recognize that the assessment of cardiovascular disease risk is important. In addition, an endocrinologist in a multidisciplinary team performs screening for metabolic diseases and extracts secondary hypertension. For this reason, our hospital routinely diagnoses whether a patient has hypertension before metabolic surgery. 

Replace the reference about the GATEWAY TRIAL (4) by the last publication: JACC VOL. 8 3, FEBRUARY13 2024: 637 – 648 

Reply：

Thank you for your kind suggestion. As you mentioned, we replaced it (Page 18, Line 335-337).

- Include 2 important references and discuss: PLoS Med 18(11): e1003817. https://doi.org/10.1371/journal.pmed.1003817 and Obes Surg 2023 Aug;33(8):2485-2492. doi: 10.1007/s11695-023-06711-2

- the PLOS Med paper stated that Sleeve gastrectomy is a factor for recurrence.

Reply：

Thank you for your constructive suggestions. The former was added to the paragraph on risk factors for HTN remission (Page 19, Line 231-237), and the latter was added to the paragraph on surgical procedures (Page 22, Line 303-304) in the section of Discussion.

Reviewer #4: The authors aimed to identify risk factors and build the prediction for HTN remission and recurrence post-LSG in severely obese patients with 62 patients. The results showed that the number of preoperative antihypertensive drugs and visceral fat area were associated and predictive of HTN remission and recurrence post-LSG.

1. Line 84. “diagnosed with HTN preoperatively and were taking oral…..” The definition of HTN should be “or”?

Reply：

As you pointed out, we changed it from “and” to “or” (Page 4, Line 81).

2. Sample recreation criteria. Is there any age restriction? Were pregnant women excluded? Please provide exclusion and inclusion criteria in the manuscript.

Reply：

Thank you for your important suggestion. Inclusion and exclusion criteria were added to the Methods section (Page 4, Line 85-87 and Page 5, Line 91-93). Specifically, the inclusion criteria are in accordance with the 2021 consensus statement of the Japanese Society for the Treatment of Obesity, Japan Diabetes Society, and Japan Society for the Study of Obesity, and a description of secondary hypertension has been added as an exclusion criterion. We would appreciate it if you could check it.

3. Why did the authors decide to present data as medians (interquartile ranges) instead of mean(SD)?

Reply：

In this study, there were several variables that did not follow a normal distribution, such as plasma renin activity and plasma aldosterone concentration, and we decided to describe the variables as median instead of mean.

4. Why was the Mann-Whitney U test used for the continuous variable? Also, the Mann-Whitney U test is a non-parametric test and is not intended for comparing the mean value.

Reply：

As mentioned above, there were several variables that did not follow a normal distribution, so we adopted the Mann-Whitney U test, which is a nonparametric test, for the comparison of values between the two cohorts. As you mentioned, the phrase “compare the mean values” is misleading, so we removed the "mean" in the manuscript (Page 6, Line 118-119). 

Sincerely,

Hideki Kumagai

Department of Surgery, School of Medicine, Iwate Medical University 

2-1-1 Idaidori, Yahaba-cho, Shiwa-gun, Iwate, 028-3695, Japan.

---

## [Decision Letter · Decision Letter 1]

29 Oct 2024

PONE-D-24-29682R1Predictive Factors Involving the Remission and Recurrence of Hypertension Post-Laparoscopic Sleeve Gastrectomy in Japanese Patients with Severe ObesityPLOS ONE

Dear Dr. Kumagai,

Thank you for submitting your manuscript to PLOS ONE. After careful consideration, we feel that it has merit but does not fully meet PLOS ONE’s publication criteria as it currently stands. Therefore, we invite you to submit a revised version of the manuscript that addresses the points raised during the review process.

We look forward to receiving your revised manuscript.

Kind regards,

Tatsuo Shimosawa, M.D., Ph.D.

Academic Editor

PLOS ONE

Journal Requirements:

Additional Editor Comments :

A minor comments are given by a reviewer.

Reviewers' comments:

Reviewer's Responses to Questions

**Comments to the Author**

1. If the authors have adequately addressed your comments raised in a previous round of review and you feel that this manuscript is now acceptable for publication, you may indicate that here to bypass the “Comments to the Author” section, enter your conflict of interest statement in the “Confidential to Editor” section, and submit your "Accept" recommendation.

Reviewer #1: All comments have been addressed

Reviewer #3: All comments have been addressed

Reviewer #4: All comments have been addressed

2. Is the manuscript technically sound, and do the data support the conclusions?

Reviewer #1: Yes

Reviewer #3: Yes

Reviewer #4: Yes

3. Has the statistical analysis been performed appropriately and rigorously? 

Reviewer #1: Yes

Reviewer #3: I Don't Know

Reviewer #4: Yes

4. Have the authors made all data underlying the findings in their manuscript fully available?

Reviewer #1: Yes

Reviewer #3: Yes

Reviewer #4: Yes

5. Is the manuscript presented in an intelligible fashion and written in standard English?

Reviewer #1: Yes

Reviewer #3: Yes

Reviewer #4: Yes

6. Review Comments to the Author

Reviewer #1: The manuscript has been revised appropriately based on the reviewer`s comments, and the research was sound.

Reviewer #3: Dear authors,

Thanks for your responses. The manuscript is much better now.

Please revise:

- In the abstract, line 22: "LSG in patients who are severely obese," change for LSG in patients who have severe obesity.

- Instead of weight regain, use the IFSO definition: Recurrent weight gain.

Reviewer #4: (No Response)

7. PLOS authors have the option to publish the peer review history of their article (what does this mean?). If published, this will include your full peer review and any attached files.

Reviewer #1: No

Reviewer #3: **Yes: **Carlos Aurelio Schiavon

Reviewer #4: No

---

## [Author Response · Author response to Decision Letter 1]

11 Nov 2024

Prof. Emily Chenette

PLOS ONE, Editor in Chief

Dear Prof. Chenette,

The comments of the reviewers have been very helpful for us to revise our manuscript. We have attempted to address the questions raised by the reviewers. Our point-by-point responses to the reviewers’ comments are provided below.

Reviewer #3: Dear authors,

Thanks for your responses. The manuscript is much better now.

Please revise:

- In the abstract, line 22: "LSG in patients who are severely obese," change for LSG in patients who have severe obesity.

Reply:

Thank you again for your kind and constructive suggestions. As you suggested, the wording has been corrected.

- Instead of weight regain, use the IFSO definition: Recurrent weight gain.

Reply：

As suggested, we have changed the word "weight regain" to "recurrent weight gain" in the manuscript in accordance with the IFSO definition.

Sincerely,

Hideki Kumagai

Department of Surgery, School of Medicine, Iwate Medical University 

2-1-1 Idaidori, Yahaba-cho, Shiwa-gun, Iwate, 028-3695, Japan.

---

## [Editor Report · Decision Letter 2]

18 Nov 2024

Predictive Factors Involving the Remission and Recurrence of Hypertension Post-Laparoscopic Sleeve Gastrectomy in Japanese Patients with Severe Obesity

PONE-D-24-29682R2

Dear Dr. Kumagai,

We’re pleased to inform you that your manuscript has been judged scientifically suitable for publication and will be formally accepted for publication once it meets all outstanding technical requirements.

Kind regards,

Tatsuo Shimosawa, M.D., Ph.D.

Academic Editor

PLOS ONE
---

## [Editor Report · Acceptance letter]

6 Dec 2024

PONE-D-24-29682R2 

PLOS ONE

Dear Dr. Kumagai, 

I'm pleased to inform you that your manuscript has been deemed suitable for publication in PLOS ONE. Congratulations! Your manuscript is now being handed over to our production team.

Kind regards, 

on behalf of

Prof. Tatsuo Shimosawa 

Academic Editor

PLOS ONE